# Congestion Control in Charging Stations Allocation with Q-Learning

**Li Zhang, Ke Gong * and Maozeng Xu**

School of Economic and Management, Chongqing Jiaotong University, Chongqing 400074, China
* Correspondence: gks_cn@163.com

**Abstract:** Navigation systems can help in allocating public charging stations to electric vehicles (EVs) with the aim of minimizing EVs' charging time by integrating sufficient data. However, the existing systems only consider their travel time and transform the allocation as a routing problem. In this paper, we involve the queuing time in stations as one part of EVs' charging time, and another part is the travel time on roads. Roads and stations are easily congested resources, and we constructed a joint-resource congestion game to describe the interaction between vehicles and resources. With a finite number of vehicles and resources, there exists a Nash equilibrium. To realize a self-adaptive allocation work, we applied the Q-learning algorithm on systems, defining sets of states and actions in our constructed environment. After being allocated one by one, vehicles concurrently requesting to be charged will be processed properly. We collected urban road network data from Chongqing city and conducted experiments. The results illustrate the proposed method can be used to solve the problem, and its convergence performance was better than the genetic algorithm. The road capacity and the number of EVs affected the initial of Q-value, and not the convergence trends.

**Keywords:** electric vehicle; public charging station; congestion game; Q-learning

---

## 1. Introduction

The zero-emission and noiseless electric vehicle (EV)—a kind of new-energy vehicle—is considered to be an effective component of a sustainable transportation system because it is environmentally friendly. However, its limited battery life makes drivers search for a proper public charging station to be charged frequently, which increases the traffic congestion of a road network. There are two ways to solve such a problem. One is to speed up the pace of construction of public charging stations [1,2], which involves consideration of the number and location of public charging stations [2–5]. The other is to utilize existing public charging stations effectively, guiding EVs to be charged with minimum time cost, avoiding congestion on roads and in stations. With the help of intelligent transportation systems (ITSs) and communication technologies, EV drivers now prefer to obey the guidance from real-time navigation systems [6]. Navigation systems can be used to allocate stations to EVs based on sufficient integrated information, such as geography, traffic congestion in a road network, and queuing situations in each charging station.

Allocating stations for EVs has been the subject of recent study. This problem has been transformed into a routing problem [7], the aim of which was to minimize EV driving time. Considering only driving time is not reasonable because of EVs' time-consuming charging processes. Generally, there are two kinds of charging patterns (i.e., fast and slow). Fast charging can charge up to 80% of a vehicle's rated battery capacity within 0.5 to 1 h, while slow charging can take 10 to 20 h for a full charge. That is, at least half an hour is required to charge a battery, even if it is done quickly [8]. In this paper, we pay more attention to typical public charging stations, which are supplied from three-phase AC mains at 50/60 Hz; under these conditions, it takes a few hours to charge a battery. EVs' charging process costs

so much time that the queuing time at stations should be deemed necessary as part of their charging time cost.

A charging process can be deemed appropriate as a congestion game [9], in which a finite number of EVs compete for a fixed number of resources, including roads and stations, in an urban area. Such a process involves interactions between EVs and their environments. EVs' charging behavior will increase the congestion of routes and stations; conversely, their levels of congestion will change EVs' charging time cost. The congestion game model, a class of non-cooperative game in which the cost of a selfish player depends on the total number of players playing the same strategy [10], can illustrate this process clearly. Players in the game can know their actions' impact on the congestible resources [11], which can be alleviated by some effective control strategy. The congestion game is a common method to solve resource allocation problems, such as communication resources [12,13], cloud computing resources [14], power resources in smart grids [15], and parking resources [13]. However, previous studies have only considered one resource at a time. Our EV charging allocation problem is a combination of routing and queuing problems, in which roads and stations are scarce and easily congested resources. Herein, we provide a joint-resource congestion game model.

Q-learning [16,17], as an off-policy temporal difference (TD) control algorithm in reinforcement learning, is a vigorous tool for Nash games and operation control [18–20], with many advantages. First, it does not need any model of the environment and is independent of the initial value, which is not true of most methods. Heuristic algorithms [14,21,22] are generated to achieve equilibrium in game theory. This means that Q-learning can be adapted to our unstable real-time traffic environment. Second, the learned action-value function in Q-learning is defined with the optimal one, independent of the policy being followed [16]. This important function ensures that the Q-value can approach its optimal value with probability 1 even under stochastic conditions on the sequence of step-size parameters, which is key to our finite EV sequence. Third, the agent in Q-learning shows a good search ability to find a global solution to multi-agent differential graphical games [23]. In our study, we first implemented the Q-learning algorithm to search the Nash equilibrium in a congestion game.

We considered a charging station allocation problem for identical EVs concurrently requesting charging in an intelligent transportation system. The navigation systems on which the Q-learning algorithm is deployed do the allocation work for these EVs one by one. One EV's allocation result is decided based on the congestion status of roads and stations which it faces, while the status is determined by its previous EV allocation result. Such continuous allocation work is a Markov decision process, and its convergence controller can be set as Bellman functions. The bottleneck in our problem is mapping the real different congestion situations into learning rewards for the Q-learning agent.

Our contribution in this paper can be summarized as follows:

(1) We extend the background congestion, defined in [24], from the average level of an area to road segments;

(2) We construct a joint-resource atomic congestion game to describe the process of EV-station allocation, considering changes in resources congestion status;

(3) We first propose deployment of the Q-learning algorithm to search for the Nash equilibrium in congestion games, transforming the resource payoffs into Q-learning rewards and control parameters.

The remainder of the paper is organized as follows. In Section 2, we present the system scenario with a congestion game model. In Section 3, we utilize the Q-learning algorithm to achieve Nash equilibrium. We present our conducted experiments and related parameter analysis of the results to show the performance of the algorithm in Sections 4 and 5. The conclusions are drawn in Section 6.

## 2. Electric Vehicle Charging Station Allocation Problem

Before presenting our problem as a game model, we discuss EV charging time cost in detail.

*2.1. EV Charging Time Cost Model*

The time cost of each EV for charging consists of the route trip time and the station queuing time, which is the total time consumed between an EV's charging request and completion.

2.1.1. Route Travel Time Cost

A route to an allocated station is a combination of some roads, and a route trip time cost is the sum of passing time costs on these roads. In an urban road network, there may be multiple routes from the recharging request location to a station.

We denote $rc_{ij}$ as the route trip time that EV *I* drives to station *j*. Generally, the value of $rc_{ij}$ should be decided by the vehicle's distance to the station, the route congestion status, and its speed. Under the condition that EVs are all kept moving at the same speed, the distance and the congestion status are the factors to be considered. In [24], the definition of the background traffic congestion was provided to distinguish congestions caused by the EVs heading for charging from other ones. However, in [24], background traffic congestion was set as the average road congestion level in one zone. To be more precise in an urban area, we redefined it at the road level because there are usually different congestion situations for different roads. In our view, a roads' congestion situation is divided into the road's general congestion status and the road's game congestion status, where the latter is caused by the charging EVs.

**Definition 1 (Road general congestion status).** *A road's normal congestion status caused by vehicles other than EVs heading for charging. Its value can be equal to the real-time road congestion indicator of Baidu, who provides an internet map service similar to Google. Let $a_k^0$ denote the general congestion status for road k.*

**Definition 2 (Road game congestion status).** *A road's congestion condition caused by EV charging activities in our scenario, described as $CO_{ik}$.*

Given a route is composed of *K* road segments, its traffic time cost can be expressed as:

$$rc_{ij} = \lambda \sum_{k=1}^{K} \left( a_k^0 + CO_{ik} \right) \times d_{ik} = \lambda \sum_{k=1}^{K} a_k^0 \times d_{ik} + \lambda \sum_{k=1}^{K} \frac{n_{ik}}{CAP_k} \times d_{ik} \tag{1}$$

which is extended from Equations (1) and (2) in [24]. Here, $\lambda$ is the coordination constant, $d_{ik}$ is the length of road *k* on which EV *i* passes, $CAP_k$ is the traffic capacity of the road *k*, and $n_{ik}$ denotes the number of EVs that have passed along this road. The congestion caused by the charging EVs on road *k* is computed as $CO_{ik} = \frac{n_{ik}}{CAP_k}$, which is determined by the value of $n_{ik}$, since the traffic capacity is a constant for a built road.

2.1.2. Queuing Time Cost

Once the station *j* is in service, EV *i* will be waiting in a queue when it is allocated to and arrives at the station. In the ITS environment, the queuing status in a station can be sensed in real time. The key problem is the time interval, which is the road trip time. The system needs to predict the queuing length in *j* at the time of *i* arriving, not at the time of *i* requesting. During the interval, there will be some EVs who leave the station after charging. The platform should collect the number of EVs in and out of the station in a timely manner.

The initial status in each station is zero vehicles, and the charging service time cst is the same for each EV. Providing that the number of EVs in the station *j* is $n_j^r$. when EV *i* requests charging, the number of EVs $n_{ij}^a$. in station *j* when EV *i* arrives can be estimated as $n_{ij}^a = n_j^r - \left| \frac{rc_{ij}}{cst} \right|$. The queuing time cost $qc_{ij}$ of EV *i* in station *j* can be approximately computed by (2):

$$qc_{ij} = cst \times n_{ij}^a. \tag{2}$$

Considering both the route trip time cost and the queuing time cost, the total time cost $C_{ij}$ of EV $i$ allocated to station $j$ can be defined as (3):

$$C_{ij} = rc_{ij} + qc_{ij}. \tag{3}$$

*2.2. Congestion Game-Based System Model*

Navigation systems managing a finite number of EVs and charging stations will do the station allocation for EVs as a private guide service. EVs will obey their guidance completely. Roads and stations are congestible resources. They can be thought of as joint resources because roads link EVs to stations. We construct a joint-resource congestion game to describe our problem.

2.2.1. Congestion Game Model

The traditional congestion game model is in the form of a four-element tuple. Based on this, we present a joint-resource atomic congestion game in the form of a combinational four-element tuple:

$$\Gamma = \left( N, (K, M), (S_i | i \in N), \left( \sum_i rc_{ik}l_{ik}, \sum_i qc_{ij}s_{ij} \right) \middle| i \in N, k \in K, j \in M \right).$$

**(1) Players**: The set $N = \{1, 2, \ldots, n\}$ denotes EVs heading for charging, and its cardinality $|N|$ represents the number of EVs.

**(2) Joint resources**: The set $M = \{1, 2, \ldots, m\}$ denotes the charging stations, and its cardinality $|M|$ represents the number of stations. The set $K = \{1, 2, \ldots, k\}$ denotes the finite number of roads which make up the traffic network for EVs heading to stations. They are both open resources shared among EVs.

**(3) Strategies**: We define $S = \{S_1, S_2, \ldots, S_i, \ldots, S_n\}$ as the strategy set of EVs. For EV $i$, the strategy is $S_i = \left\{ \left( \bigcup_{k \in K} l_{ik}, s_{ij} \right) \middle| i \in N, j \in M, k \in K \right\}$, where $\bigcup_{k \in K} l_{ik}$ tracks the roads and $s_{ij}$ records the station to which it is allocated. This is a singleton congestion game since each EV will be charged at a station. We can deduce that for each EV $i$, there exists $s_{ij} = \begin{cases} 1 & station\ j\ is\ selected\ to\ EV\ i \\ 0 & others \end{cases}$, $l_{ik} = \begin{cases} 1 & road\ k\ is\ selected\ to\ EV\ i \\ 0 & others \end{cases}$, and $\sum_j s_{ij} = 1$. In total, we can derive that there should be $\sum_i \sum_j s_{ij} = |N|$.

We also set a status matrix from the stations' view, which is $\eta = (\eta_1, \eta_2, \ldots, \eta_m)$ and $\eta_j = \left| \left\{ s_{ij} = 1 \middle| j \in M \right\} \right|$, to watch the congestion status of each station.

**(4) Payoff**: $\left\{ \left( \sum_i rc_{ik}l_{ik}, \sum_i qc_{ij}s_{ij} \right) \middle| i \in N, k \in K, j \in M \right\}$ denotes the costs of congested resources (i.e., roads and stations), varying according to the number of EVs allocated to them.

We hope that the platform allocates stations to EVs for minimizing their own time costs. This can be expressed as in (4):

$$
\begin{aligned}
k, \mathrm{m} = \mathrm{argmin}&\left( \sum_k rc_{ik}l_{ik} + \sum_j qc_{ij}s_{ij} \right), \\
\mathrm{s.t}\ &s_{ij} \in \{0, 1\}, \\
&\sum_j s_{ij} = 1, \\
&\sum_i \sum_j s_{ij} = |N|
\end{aligned}
\tag{4}
$$

2.2.2. Existence of Nash Equilibrium

In our problem, there is a finite number of charging stations, roads, and EVs. According to (1), as the number of EVs choosing the same station and roads increases, the roads and station will be more congested and their usage costs increase incrementally. The navigation system knows the strategy for each EV. Once an EV find its suitable route and station, the system achieves a temporary equilibrium

status to deal with the later EV until all EVs are allocated. No EV can decrease its cost by unilaterally changing its own strategy. This is the Nash equilibrium in the congestion game, which is expressed by (5):

$$c_i\left(s_i^*, s_{-i}^*\right) \le c_i\left(s_i, s_{-i}^*\right), \ i \in N, \ s_i, s_{-i}, s_i^* \in S^t. \tag{5}$$

In this Equation, $s_i^*$ is the optimized strategy vector of EV *i*. $s_{-i}^*$ denotes the strategy vector profile of players except for EV *i*.

## 3. EV Station Allocation Based on Q-Learning Algorithm

The joint-resource congestion game can describe the allocation process clearly, accompanied with the interaction between EVs and resources. In a road-level traffic status consideration, the complexity of an urban road network will increase the difficulty in searching. We know that the system will do the allocation work for EVs one by one, while the result is determined by the status of roads and stations to which the EV faces. Q-learning, an incremental method for dynamic programming [17], is considered appropriate for such a situation.

Q-learning is an agent-based method in which the agent interacts with its environment and adjusts its actions based on stimuli received in response to its actions [25]. There are three basic elements in the algorithm: environment, state, and action. We will introduce the algorithm after setting the elements.

### 3.1. Environment, State, and Action Set

The environment is a fundamental element in Q-learning, in which the agent chooses its actions according to corresponding rewards. In our scene, according to the joint-resource congestion game model, the environment should involve roads between EVs and those optional stations with their length and traffic status. We construct a grid world whose unit is determined by the shortest road length, and deploy resources in the grids according to their relative distances. If a road is not an integer multiple of the unit, it will cross a number of grids. There should be some grids containing segments of two joint roads. The payoff of each grid is initialized as the road's general congestion status. For those grids containing mixed roads, if their general congestion statues are different, the grid's initial payoff is set as the higher value. The accessibility of roads can also be shown in the grids by setting its initial payoff value as a large value.

The state set in our scene is to make the position of the agent visible. Each grid is a state, and the state set can be denoted as state $= \{1, 2, \ldots, s\}$, where s is the total number of grids. For each grid, it has an incremental reward once it is on the route the agent chooses, whose value is determined by its road game congestion status. In this way, the system records the agent's accumulated reward from the environment and its action effect on the environment. The action set for the agent in this grid world denotes the way the agent changes its state. In the grid world, the agent can move up, down, left, and right. That is, the action set can be denoted as Action $= \{\text{up}, \text{down}, \text{left}, \text{right}\}$.

### 3.2. Q-Learning Algorithm

The Q-learning algorithm is based on an action–value function. It has two input parameters: state and action. Our aim was to minimize the time cost. Generally, the update of the state–action function (the controller function) is realized by the Bellman equation, with the temporal difference method shown in (6):

$$Q_{t+1}(s, a) = (1 - \alpha)Q_t(s, a) + \alpha(r(s, a) + \gamma \min_{a'} Q_t(s', a')) \tag{6}$$

where $\alpha \in [0, 1]$ is the learning rate, $\gamma \in [0, 1]$ is the discounting factor, $r(s, a)$ is the immediate reward, and $Q_t(s, a)$ is the Q-value at time t.

For each EV, there is a learning process. The reward of each grid should be updated once an EV finishes its learning. All possible state–value pairs should be tested. We use an $\varepsilon - $ greedy policy in the learning process to improve learning efficiency. For each EV, with this policy, the agent will

choose a random action with $\varepsilon$ probability and an action greedily to the minimized Q-value with $1 - \varepsilon$ probability. Details can be seen in Algorithm 1.

---

**Algorithm 1**: Q-learning algorithm

---

**Input**: $N$—The Number Of Evs
$\qquad$ $M$—The Number Of Stations
$\qquad$ $K$—The Number Of Roads
$\qquad$ *Epsilon*—$E \in [0,1]$, Explore Factor, A Designed Constant
$\qquad$ $\lambda$, $CAP$, $\beta$—Designed Constant
$\qquad$ $D$—The Distance Matrix
$\qquad$ $A^0$—The Roads General Congestion Matrix
**Output**: *History*—The Allocation Strategy

---

Initialization:
$\quad$ ① $(S, A)$
$\quad$ ② *rewards* with roadsaverage congestion factor'
$\quad$ ③ $History = \begin{bmatrix} \underbrace{0, 0, \ldots 0}_{m} \end{bmatrix}$
Set the terminalSet as the three stations' positions
Repeat (for each EV):
$\quad$ Repeat for the episode:):
$\qquad$ Choose status and action $(S, A)$ using policy $\varepsilon$-greedy
$\qquad\quad$ Repeat (for each step of the episode):
$\qquad$ Take action $A$, observe $R, S'$
$\qquad\quad$ Choose $A'$ from $S'$ using policy $\varepsilon$-greedy
$\qquad\quad$ Update status–activity with Bellman equation
$\qquad\quad$ $S \leftarrow S'$; $A \leftarrow A'$.
$\qquad\qquad$ Update *rewards* for passing roads and the selected station
$\qquad$ Until $S$ is in terminalSet
$\quad$ Update the output *History*

---

## 4. Experiments and Results

All the following experiments were performed by Python 3.6 on Windows Server 2008 R2 Enterprise, 64bits, Intel(R) Xeon(R) CPU E5-2609 1.90 GHz, RAM 256 G.

To validate the proposed method, we collected the geographic data of roads from a real urban area in Chongqing, a city in the southwest of China, whose Baidu Map screenshot showing in Figure 1, and conducted the experiment using the method described above.

### 4.1. Data Introduction

In Figure 1, the three red circles with numbers denote the optional charging stations. For simplicity, we supposed a concurrent charging request happening at the same place where the circle is labeled with "S". We considered three routes—one for each station. According to the Baidu Map, the distance ratio between S and the three stations was 6:3:4. Each grid represents one distance unit. We formed the grid world as shown in Figure 2a, in which the relative positions of the three stations to the requesting location can be expressed. Roads from "S" to each station are set as lines in Figure 2a according to their distance ratio. We obtained the real general congestion status by the congestion indicator. The general congestion situations on these roads were set as the initial reward for related grids, as shown in Figure 2b. The data show that the most congested road segment was B1, while the least congested one was S3, where B is one cross point on the way to Station 1. For those grids in which optional stations were deployed, the initial statuses were set as 1 to 10,000.

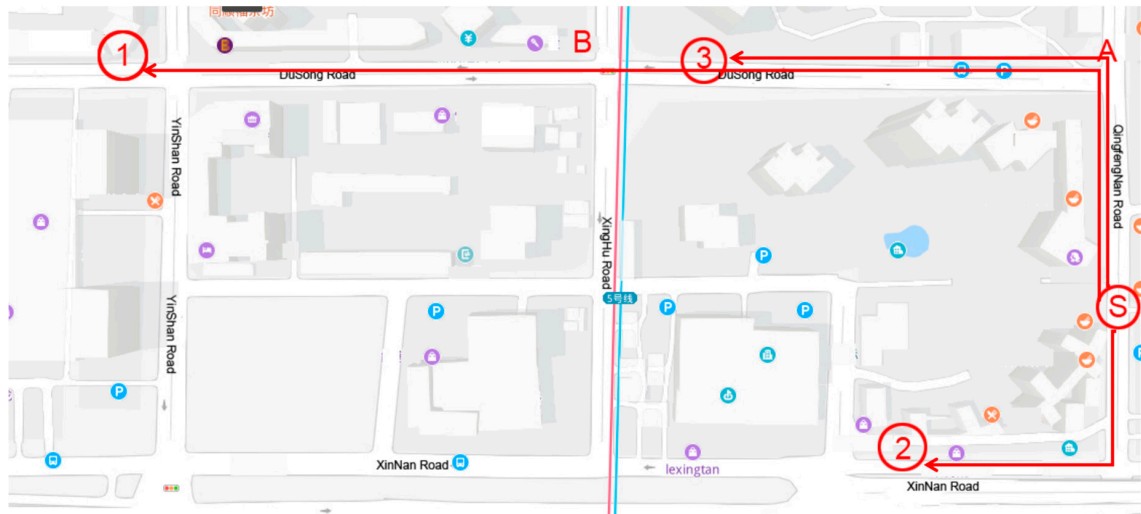

**Figure 1.** Experimental scenario: an example area in Chongqing City, China.

In the grid world in Figure 2a, each grid signifies one state. There were a total of 36 states in our system. The terminal state was one of the three optional stations deployed in grids 11, 14, and 18, with incremental reward 1. The final terminal status was decided by the minimum reward from "S" to the three optional statuses.

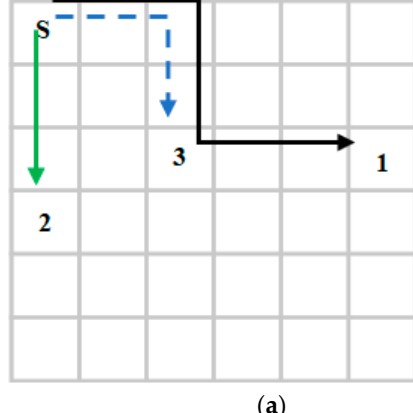

(a)

| 10,000 | 0.01 | 0.01 | 10,000 | 10,000 | 10,000 |
| 0.1 | 10,000 | 0.01 | 10,000 | 10,000 | 10,000 |
| 0.1 | 10,000 | 1 | 0.5 | 0.5 | 1 |
| 1 | 10,000 | 10,000 | 10,000 | 10,000 | 10,000 |
| 10,000 | 10,000 | 10,000 | 10,000 | 10,000 | 10,000 |
| 10,000 | 10,000 | 10,000 | 10,000 | 10,000 | 10,000 |

(b)

**Figure 2.** The grid world with initial rewards. (**a**) The real situation's map into a grid world; (**b**) The initial reward for each grid.

### 4.2. Experiments and Results Statement

We set $\lambda = 1$, cst $= 1$, CAP $= 3$, epsilon $= 0.8$, episode $= 1000$ in the allocation experiment for EVs. In the beginning, we supposed that there were no EVs in stations. The agent learned in the environment as shown in Figure 2.

#### 4.2.1. Experiment for One EV

Figure 3 illustrates the Q-value convergence performance of Q-learning for one EV. The agent chose Station 3 as the target station for this EV, which was the least congested one. From the curve, it appears to have converged when the number of iterations was almost 200. According to (1), we know that time costs from S to the three options were 1.03, 0.2, and 0.03, respectively. The agent chose the cheapest one.

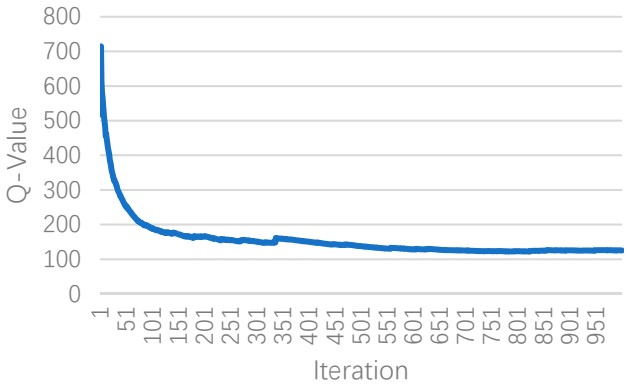

**Figure 3.** Q-value convergence performance for one electric vehicle (EV).

### 4.2.2. Experiment for 20 EVs

For each EV, the agent learned to find the right station for one EV in 400 episodes. Table 1 illustrates the allocation results in the first 12 simulations. The numbers in the station columns are the final EV allocation numbers corresponding to the stations. The results show that the agent could finish the continuous allocation task correctly. After simulating 100 times with our initial setting, the agent assigned the most EVs to Station 3. In Table 1, there was a probability of 83.3% that the number of vehicles allocated to Station 2 was greater than or equal to that of Station 1. We know that the roads' average congestion status satisfied $a_{s1}^0 > a_{s2}^0 > a_{s3}^0$ The results show that the road congestion situation lowered the allocation possibility to its related station.

**Table 1.** Allocation results in the first 12 simulation rounds.

| Iteration | Station 1 | Station 2 | Station 3 | Iteration | Station 1 | Station 2 | Station 3 |
|:---:|:---:|:---:|:---:|:---:|:---:|:---:|:---:|
| 1 | 3 | 5 | 12 | 7 | 1 | 8 | 11 |
| 2 | 3 | 7 | 10 | 8 | 1 | 1 | 18 |
| 3 | 5 | 7 | 8 | 9 | 6 | 6 | 8 |
| 4 | 3 | 7 | 10 | 10 | 6 | 6 | 8 |
| 5 | 5 | 6 | 9 | 11 | 3 | 7 | 10 |
| 6 | 6 | 2 | 12 | 12 | 4 | 3 | 13 |

### 4.3. Comparison of Q-Learning and Genetic Algorithms

The genetic algorithm (GA) is used to solve multi-objective resource allocation problems (RAPs) [26]. We chose the GA as one baseline to run a selection simulation for one EV, for which the crossover rate was 0.85, the mutation rate was 0.01, population size was 3, and we obtained the convergence performance shown in Figure 4. The curve in Figure 3 is steeper with a clear convergence trend compared to the one in Figure 4.

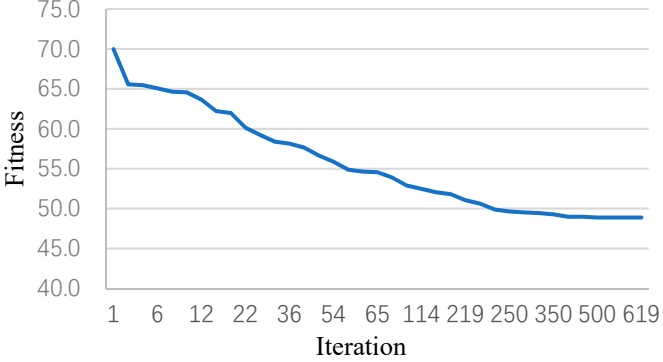

**Figure 4.** Genetic algorithm (GA) convergence performance for one EV.

## 5. Discussion

Our case study was in a fixed settled environment. The congestion status in the road network was changing, while the allocation process for each EV was the same. To the best of our knowledge, this is the first time the reinforcement learning method has been deployed to do such an adaptable allocation. Compared to the genetic algorithm, we found that our method had better convergence performance in our problem. For further consideration, other parameters (e.g., the road capacity and the number of EVs) should be tested for their effects on the convergence performance.

### 5.1. Parameters Sensitivity Analysis

#### 5.1.1. Road Capacity

The road capacity CAP is the key parameter, defined as the maximum traffic flow obtainable on a given road using all available lanes—the smaller the CAP value, the narrower the road. The roads' congestion status will change once EVs change their strategies. We changed the value of CAP and observed its effects on the system's convergence performance.

In Figure 5a, the three curves represent three kinds of road capacity—1.25, 3, and 5. The convergence trends in these three curves are the same. However, there are relatively large differences in the initial Q-Value. The smaller the CAP, the smaller the initial Q-Value.

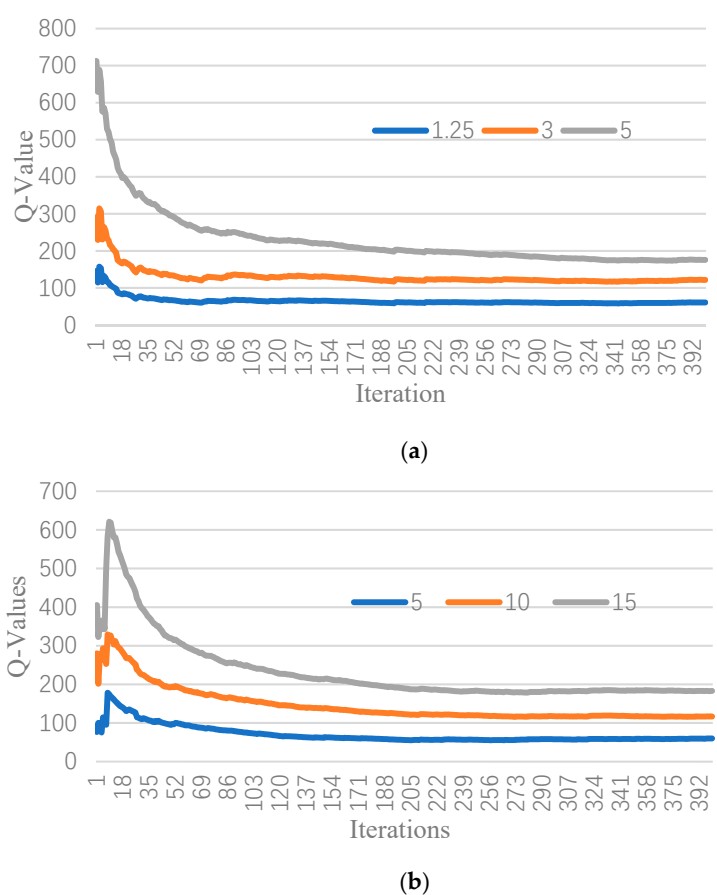

**Figure 5.** Analysis of parameters affecting convergence: (**a**) the road capacity; (**b**) the number of EVs.

#### 5.1.2. Number of EVs

Figure 5b illustrates the relationship between the number of EVs concurrently requesting and the convergence performance. According to the figure, for a single EV, convergence was achieved quickly.

However, with increasing numbers of EVs, there was a slight difference in convergence, since the resources were all congested.

## 6. Conclusions

In this paper, we investigated a strategy for the allocation of EVs to stations with a reinforcement learning Q-learning algorithm, which can be deployed on navigation systems. We considered time costs both on roads and at stations, which were affected by their congestion status. We used a grid world to set the simulation environment. The target stations were fixed in special grids according to their distance from the start point. Each grid had its own reward, which was mapped as the time costs of using its corresponding road or station. The action set included four elements: up, down, left, and right. The centrally managed cloud platform allocated EVs one by one. The terminal allocation result for each EV was the station minimizing its sum reward. Each EV's strategy was determined by the environment in which it was. This was treated as a Markov decision process (MDP). The experiments' results indicated that the Q-learning algorithm could do the allocation work intelligently by considering the congestion status of roads and stations. Q-learning could achieve better convergence performance than a genetic algorithm. The road capacity and the number of EVs both affected the initial Q-value, while the convergence trends were the same. Further study will extend the simulation to distributed start positions with a speedy reinforcement learning algorithm.

**Author Contributions:** Original draft preparation, L.Z.; review, K.G. and M.X.; supervision, K.G. and M.X.; validation, L.Z. and K.G.

**Funding:** This research was funded by National Science Foundation of China, grant number 71471024 and 71871034; Science and Technology Research Program of Chongqing Municipal Education Commission, grant number KJ1705119; Basic science and frontier technology of Chongqing Science and Technology Commission, grant number cstc2017jcyjA1695; China Postdoctoral Science Foundation, grant number 2014M560711 and 2015T80974.

**Conflicts of Interest:** The authors declare no conflict of interest.

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
