# Peer review of "Congestion Control in Charging Stations Allocation with Q-Learning"

_sustainability, doi:10.3390/su11143900_

Round 1

Reviewer 1 Report

The authors study the charging station allocation problem for EVs, with explicit consideration of traffic congestion. In order to realize a self-adaptive allocation, the authors used a reinforcement learning approach (Q-learning). They performed numerical experiments using geographical data based on the real urban area of Chongqing (southwest of China). 

Searching the Nash equilibrium in a congestion game based on reinforcement learning, more specifically, Q-Learning algorithm is important and relevant for the Sustainability field. So, the paper in its current form requires some revision.  

From my perspective, one issue – methodology design – has not been discussed in a convincing way, namely the algorithmic part regarding the choice of the Q-Learning algorithm is not well justified. In fact, the choice of the reinforcement learning algorithm is not well explained. Why Q-Learning algorithm? For reinforcement learning algorithms, in addition to Q-Leaning, many efficient algorithms (Monte Carlo method, among others) have been introduced. I recommend the authors to see the following book:

-       Sutton, Richard S., and Andrew G. Barto. Reinforcement learning: An introduction. MIT press, 2018.

Finally, the title may be enhanced as follows: Congestion Control in Charging Station Allocation by Application of Reinforcement learning

Sincerely,

Author Response

Response to reviewer 1’ comments

Point 1: From my perspective, one issue – methodology design – has not been discussed in a convincing way, namely the algorithmic part regarding the choice of the Q-Learning algorithm is not well justified. In fact, the choice of the reinforcement learning algorithm is not well explained. Why the Q-Learning algorithm? For reinforcement learning algorithms, in addition to Q-Leaning, many efficient algorithms (Monte Carlo method, among others) have been introduced. I recommend the authors to see the following book: Sutton, Richard S., and Andrew G. Barto. Reinforcement learning: An introduction. MIT Press, 2018.

Response 1: We utilize the Q-learning algorithm to propose a model to solve our problem because it has advantages which make it suitable for solving our problem.

First, Q-learning is an off-policy temporal-difference control algorithm, which does not need any model of the environment and is independent of the initial value. This makes it can be adapted to our unstable real-time traffic environment.

Second, the learned action-value function in Q-learning is defined with the optimal one, independent of the policy being followed. This makes it can find optimal behavior generally in one step learning. Furthermore, under stochastic conditions on the sequence of step-size parameters, Q-value can approach to its optimal value with probability 1. In our problem, there is a finite number of EVs, whose parameters produce one sequence. Q-learning can do a good performance in solving this problem.

Third, we find Q-learning has better convergence speed than the genetic algorithm in our experiment. Quick convergence can be helpful in the real-time system.

Finally, Q-learning shows good search ability in finding a global solution to multi-agent differential graphic games in reference paper 23. Even EVs will be allocated in sequence, we still hope a global ideal solution to all of them.

Monte Carlo methods do not require a model of environment either. However, it is based on the average reward which makes the method can be incremental based on an episode, not a step. This also affects their difference in convergence speed which is shown in figure 6.7 in the recommended book. Once the number of EVs increases, the advantage of Q-learning can be manifest.

Accordingly, we add some evidence in the introduction and reference to support the formulation of our proposed model.

Point 2: Finally, the title may be enhanced as follows: Congestion Control in Charging Station Allocation by Application of Reinforcement learning

Response 2: Thanks for your constructive suggestion. We only considered the Q-learning algorithm here. If we use ‘Reinforcement learning’ in the title, many other reinforcement learning methods should be studied in our problem to support it. The current content is not enough to meet the requirements of this enhanced title. It is the direction of our further research. 

Reviewer 2 Report

This is very interesting paper and I’m very glad that I was the possibility to review it. The structure is appropriate. The problem is clear identified. The methods, analysis and results are well described. However, it’s important to emphasize some remarks:

Utilization of Q-learning algorithm is interesting. Nevertheless, the Authors should describe in more details why they chose this method and what is the advantages in the confirmation to the others?

It is not necessary to add the equations in the text (lines 119, 124, 135, 174). The number is enough. The equation should be mentioned in separate line only.

There are some technical mistakes related to the references (line 231-232, 304).

Author Response

Response to reviewer 2’ comments

Point 1: Utilization of the Q-learning algorithm is interesting. Nevertheless, the Authors should describe in more details why they chose this method and what is the advantages in the confirmation to the others?

Response 1: We utilize the Q-learning algorithm to propose a model to solve our problem because it has advantages which make it suitable for solving our problem.

First, Q-learning is an off-policy temporal-difference control algorithm, which does not need any model of the environment and is independent of the initial value. This makes it can be adapted to our unstable real-time traffic environment.

Second, the learned action-value function in Q-learning is defined with the optimal one, independent of the policy being followed. This makes it can find optimal behavior generally in one step learning. Furthermore, under stochastic conditions on the sequence of step-size parameters, Q-value can approach to its optimal value with probability 1. In our problem, there is a finite number of EVs, whose parameters produce one sequence. Q-learning can do a good performance in solving this problem.

Third, we find Q-learning has better convergence speed than the genetic algorithm in our experiment. Quick convergence can be helpful in the real-time system.

Finally, Q-learning shows good search ability in finding a global solution to multi-agent differential graphic games in reference paper 23. Even EVs will be allocated in sequence, we still hope a global ideal solution to all of them.

Monte Carlo methods do not require a model of environment either. However, it is based on the average reward which makes the method can be incremental based on an episode, not a step. This also affects their difference in convergence speed which is shown in figure 6.7 in the recommended book. Once the number of EVs increases, the advantage of Q-learning can be manifest.

Accordingly, we add some evidence in the introduction and reference to support the formulation of our proposed model.

Point 2: It is not necessary to add the equations in the text (lines 119, 124, 135, 174). The number is enough. The equation should be mentioned in separate line only.

Response 2: Thanks for your advice. In this revision, we delete the equations in the text and use the number to represent them. Details can be seen in the document named ‘Trace version’.

Point 3: There are some technical mistakes related to the references (line 231-232, 304).

Response 3: Thanks for your careful inspection. We check and update the cross-references in this article.
